# Traffic Inequality and Relations in Maritime Silk Road: A Network Flow Analysis

**Naixia Mou** [1], **Haonan Ren** [1], **Yunhao Zheng** [1], **Jinhai Chen** [2,*], **Jiqiang Niu** [3], **Tengfei Yang** [4], **Lingxian Zhang** [1] **and Feng Liu** [1]

1   College of Geodesy and Geomatics, Shandong University of Science and Technology, Qingdao 266590, China; mounx@lreis.ac.cn (N.M.); 201882020059@sdust.edu.cn (H.R.); 201882020061@sdust.edu.cn (Y.Z.); 993851@sdust.edu.cn (L.Z.); luf3286@sdust.edu.cn (F.L.)
2   National-Local Joint Engineering Research Center for Marine Navigation Aids Services, Navigation College, Jimei University, Xiamen 361021, China
3   Key Laboratory for Synergistic Prevention of Water and Soil Environmental Pollution, Xinyang Normal University, Xinyang 464099, China; niujiqiang@xynu.edu.cn
4   Aerospace Information Research Institute, Chinese Academy of Sciences, Beijing 100094, China; yangtf@radi.ac.cn
*   Correspondence: jhchen@jmu.edu.cn

**Abstract:** Maritime traffic can reflect the diverse and complex relations between countries and regions, such as economic trade and geopolitics. Based on the AIS (Automatic Identification System) trajectory data of ships, this study constructs the Maritime Silk Road traffic network. In this study, we used a complex network theory along with social network analysis and network flow analysis to analyze the spatial distribution characteristics of maritime traffic flow of the Maritime Silk Road; further, we empirically demonstrate the traffic inequality in the route. On this basis, we explore the role of the country in the maritime traffic system and the resulting traffic relations. There are three main results of this study. (1) The inequality in the maritime traffic of the Maritime Silk Road has led to obvious regional differences. Europe, west Asia, northeast Asia, and southeast Asia are the dominant regions of the Maritime Silk Road. (2) Different countries play different maritime traffic roles. Italy, Singapore, and China are the core countries in the maritime traffic network of the Maritime Silk Road; Greece, Turkey, Cyprus, Lebanon, and Israel have built a structure of maritime traffic flow in the eastern Mediterranean Sea, and Saudi Arabia serves as a bridge for maritime trade between Asia and Europe. (3) The maritime traffic relations show the characteristics of regionalization; countries in west Asia and the European Mediterranean region are clearly polarized, and competition–synergy relations have become the main form of maritime traffic relations among the countries in the dominant regions. Our results can provide a scientific reference for the coordinated development of regional shipping, improvement of maritime competition, cooperation strategies for countries, and adjustments in the organizational structure of ports along the Maritime Silk Road.

**Keywords:** traffic flow; traffic inequality; traffic relations; Maritime Silk Road; network flow analysis

## 1. Introduction

Traffic promotes resource flow, which leads to globalization, spatial differentiation of social and economic activities, and expansion of regional traffic networks [1]. Therefore, traffic plays a guiding, supporting, and guaranteeing role in regional and national development and is an important parameter that reflects spatial relations in trade [2]. The traffic flow generated by various transport modes is an important space flow, which can deeply reflect the interregional and international economic, trade, and political relations [3]. World maritime trade accounts for more than 80% of the total world merchandise trade [4], and can greatly affect the world economy [5]. Therefore, it can be said that the shipping industry is a barometer of international economic trends.

With the continuous advancement of the Maritime Silk Road initiative, the economic and trade exchanges between the regions along the route have become closer, and the maritime traffic contacts have become more frequent; this has injected more and more energy into world trade cooperation [6]. However, the political, economic, trade, and resource environments of the countries/regions along the Maritime Silk Road are different, which creates the phenomenon of traffic inequality. As a result, the connections between countries/regions and their respective roles are significantly different. Turning this inequality into a development opportunity, using the greatest advantages of the Maritime Silk Road initiative, and making economic and trade cooperation between countries/regions along the route more reasonable and efficient [7], have always been topics of great concern to governments and societies. In this context, analyzing the traffic inequality and relations in the Maritime Silk Road is important for gaining insights into the weaknesses of economic and trade cooperation and seizing development opportunities.

Existing studies on traffic inequality mostly focus on port performance [5,8], port concentration [9,10], hub port position [11] and other port characteristics. In fact, the roles that countries play in the maritime network and the traffic relations formed with other countries are also important reflections of traffic inequality. The exploration of these roles will help countries coordinate maritime trade relations from the national strategic perspective, formulate macro and efficient economic and trade cooperation strategies, strengthen the construction of maritime power, and provide scientific references for realizing regional economic integration [12]. Unfortunately, there is almost no relevant literature to explore traffic relations at the national level from the perspective of traffic inequality. In the maritime network, traffic inequality and relations are determined by the density and direction of meaningful links. A network flow analysis provides a theoretical foundation to explore the position and relations of various nodes in the network [13]. In addition, the construction of maritime traffic networks and a high-quality analysis of the network flow are inseparable from a large amount of accurate ship movement trajectory data. Studies have shown that high temporal-spatial resolution data provide detailed trajectories. The data analysis results have great potential to support policies and have become a necessary prerequisite for current traffic flow network analysis [14]. Automatic identification system (AIS) sensors can obtain real-time dynamic and static information along with ship navigation information [15–17]. In recent years, the development of AIS has promoted traffic flow studies [18], which provide reliable research data for exploring the lesser known attributes of maritime traffic such as marine space transportation characteristics and maritime traffic flow patterns.

In this study, we construct the Maritime Silk Road's traffic network based on the AIS trajectory data; additionally, we investigate traffic inequality and relations using complex network theories, social network analysis, and network flow analysis. The structure of this article is as follows: Section 2 introduces related studies on traffic inequality along with traffic status and relations of the maritime network, Section 3 introduces the methods that support this research, and Section 4 presents the empirical results. The last two sections analyze and discuss the results in depth and draw conclusions about traffic inequality and relations in the Maritime Silk Road.

## 2. Literature Review

### 2.1. Traffic Inequality

Traffic inequality is a broad concept. There are significant differences in the political background, economic environment, and the resource conditions in various regions; their respective traffic needs are different, thus presenting a state of traffic inequality. With traffic inequality as the background, researchers have made extensive explorations of urban traffic services [19,20], environmental pollution [21,22], and social problems [23,24].

Of course, there is no lack of research on maritime traffic inequality, and most of them are carried out at two scales, port scale and regional scale. At the port scale, the existing research on traffic inequality is primarily focused on cargo concentration, hierarchical

structure, and spatial interaction. For example, Notteboom [25] used the Gini coefficient to compare the evolution of cargo-flow concentration in the European and the North American ports from 1975 to 2003 and implemented the Gini coefficient decomposition analysis method proposed by Dagum [26] to extract more detailed spatial dynamic information than that obtained from traditional methods. A comparison in the ability of the two ports to control traffic flow indicates that the North American ports have a higher degree of cargo concentration, and some of these port areas have come to occupy the dominant position in the entire system. Wang et al. [27] explored the spatial pattern of traffic inequality based on graph theory, selected the dominant east Asian ports as research cases, analyzed their hierarchical structure and spatial interaction, and assessed the impact of traffic consolidation on the container industry. At the regional scale, research on traffic inequality is mainly reflected in the role of the region. For example, Xu et al. [28] used a social network analysis and a dominant flow analysis to investigate the unequal evolution of the status of various shipping areas in the global shipping network and found that East Asia is superior to other regions in total traffic flow; however, its position is lower than that of northwestern Europe and the Mediterranean region in Europe, which proves the hypothesis that the regional maritime position does not fully reflect the total traffic volume due to the complexity of the maritime traffic network. Although the existing studies have provided comprehensive research at the port and regional levels, there is still a lack of relevant research on the specific area of the Maritime Silk Road and the national level analysis of maritime traffic relations from the perspective of traffic inequality.

At present, research methods related to maritime traffic inequality are mainly divided into two categories, an index comparison method and a network flow analysis method. The first type mainly consists of descriptive indicators, such as the Gini coefficient [29], Lorenz curve [30], and Herfindahl–Hirschman Index [31]. Their common feature is their poor interpretability and that they can only be measured by comparing different index values [32]. They do not have the ability to explore the dynamic information behind the traffic inequality mode. The second type of method uses the connection properties of the network to find the dominant nodes in the network based on the intensity and flow direction of the traffic and analyzes the spatial dynamic information of the traffic inequality mode. Comparing the two types of methods, the latter method, which has irreplaceable advantages in the determination and maintenance of relationships between individuals, not only considers geographic space attributes but also has the ability to explore traffic flow pattern formation and operation mechanisms in a more comprehensive and detailed way.

### 2.2. Traffic Relations and Status

With the continuous promotion of the 21st-Century Maritime Silk Road initiative, studies on the shipping industry with the Maritime Silk Road chosen as a specific area have become more and more abundant. Li et al. [33] developed an evaluation system based on the literature and the analytic hierarchy process (AHP) (an entropy method), which analyzed the evolution of commercial maritime power in 32 countries/regions since the beginning of the Belt and Road initiative and found that Vietnam (instead of China) has the highest average annual growth rate. Zhao et al. [34] studied the energy interdependence between China and countries along the Belt and Road region and found that they have established an interdependent relationship; it was also observed that China is in a passive position relative to the countries located along the route. Yu et al. [35] used a multilevel spatiotemporal dynamics framework to analyze the temporal and spatial dynamics of the global ocean network and found that China, Singapore, South Korea, and other ports in the countries located along the Silk Road have established new shipping relationships; the new connections carry significant traffic flow. However, most of the existing research on shipping relations remains at the level of studying the breadth and intensity of the connections and does not go deeply into the quantitative study of relation types.

As is well known, shipping constitutes a complex network. Many scholars have used a complex network theory to conduct in-depth research on shipping relations. For

example, Kitamura et al. [36], Pais Montes et al. [37], and Mou et al. [38] used the complex network theory to analyze the competition and cooperation relations, spatial association modes, and route evolution among countries in the shipping network for different cargo types. The complex network theory calculates and analyzes all the edges in a large-scale network; however, not all edges are meaningful. Additionally, many insignificant edges affect the analysis of important relationships [39]. Therefore, traffic flow analysis assumes that the relations between any pair of nodes depends on the direction and intensity of the meaningful traffic flow between them. The volume of the traffic flow calculated is greater than the given threshold. A change in any connection or any elimination will affect the spatial configuration of the entire network [27]. Cullinane et al. [40] used the multiple linkage analysis method in network flow theory to demonstrate that the importance of a port and its spatial interaction with other ports have a large correlation with the amount and intensity of traffic flow in and out of the port. Network flow analysis provides direction for the study of major relations in shipping networks [41]. At present, on the basis of network flow theory, the spatial distribution of significant traffic flow among countries is often explored while the relative importance of countries and their important interrelations are rarely studied.

In addition, in studies regarding the status of shipping network nodes, early scholars, such as Jiang and Jiang et al. [42], Yap et al. [43], and Low et al. [44] used port throughput, which measures the amount of cargo that the port can handle, as a measure of the strength of a port's role in the maritime network. However, with further studies, scholars have gradually realized that the network is an interactive and complex system, and that the status of nodes depends more on their relationship position in the network rather than a certain index value [45]. Ducruet et al. [13], using the status of the northeast Asian ports in the maritime network as a research case, compared graph theory and network analysis methods with traditional measurement methods that use throughput as an indicator and empirically demonstrated that centrality, connectivity, and vulnerability indicators have more advantages than throughput in describing the status and role of nodes. Among them, centrality is an important concept in social network analysis theory. In fact, many scholars have adopted social network analysis to study the status of nodes in a network. For example, Song et al. [46] used a social network analysis to study the structure of a linear transportation network centered on a sub-hub port of Gwangyang in South Korea and found that Busan is the most important port in the transport network in terms of the number of connections. However, when considering the hub centrality, the influence of the Shanghai and Hong Kong ports is more significant. Li et al. [47] divided global shipping into 25 shipping areas and analyzed the evolution of the status of each shipping area in the global shipping network during the period 2001–2012 using centrality indicators. They concluded that Europe had been the center of global shipping network to a large extent during the study period; however, its central position was gradually declining, and the global shipping center was moving eastward.

Therefore, on the basis of relevant literature reviews, we use the AIS trajectory data of ships to extract the OD (origin–destination) traffic flow, build a Maritime Silk Road traffic network, and use the complex network theory along with social network analysis and dominant flow analysis methods to detect inequality in the Maritime Silk Road traffic. We also extend our study to identify the dominant areas along the route, and we adopt a multiple linkage analysis method to achieve a more detailed study of the role and relations of maritime traffic among the countries located in dominant regions.

## 3. Materials and Methods

### 3.1. Study Area

There are two major routes of the Maritime Silk Road that form our study area: (i) from the Chinese coastal ports across the South China Sea, via the Malacca Strait and the Indian Ocean to reach Europe, and (ii) from the Chinese coastal ports to the South Pacific Ocean. The study area consist of 64 countries, including northeast Asia, southeast Asia,

Oceania, south Asia, west Asia, northeast Africa, east Africa, south Africa, and Europe, as shown in Figure 1.

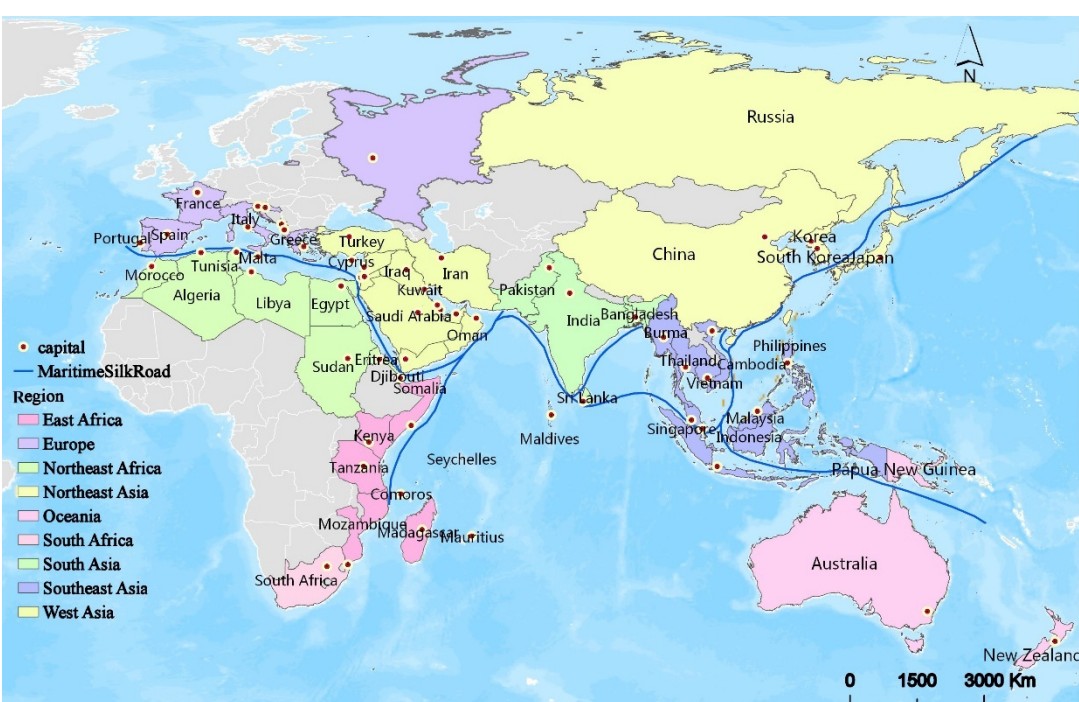

**Figure 1.** Study area routes (given in blue lines) and Maritime Silk Road countries. The study area consist of 9 regions.

### 3.2. Data

Ship arrival and departure information is extracted from the real-time ship dynamic records provided by the AIS trajectory data and the world port index (WPI) data, and each ship departure and arrival is used as the OD data. Ultimately, 257,715 traffic flow OD data are obtained.

One can use the real-time latitude and longitude information contained in the AIS data to locate the navigation routes of ships 1, 2, 3, 4, and 5, as shown in Figure 2a. Time-series locations between ports can be viewed as a trajectory for each vessel. For example, ship 1 has trajectories between ports AB, BC, CD, DE, EF, and FB; ship 2 has trajectories between ports AD, DE, EF, FC, and CB; ship 3 has trajectories between ports AF, FC, CD, and DE; ship 5 has trajectories between ports BA, AE, and ED. The marine network between ports is created by connecting each port pair in the track as a link (as shown in Figure 2b). Combining the WPI and the national basic information data, the Maritime Silk Road ports are divided into countries, and the arrival and departure records of each country's ports are added together to form a maritime traffic network (with countries as the nodes). The network has at least one connecting edge with a weight of the total maritime traffic between the two countries, resulting in a maritime traffic network of 64 countries along the Maritime Silk Road (as shown in Figure 2c).

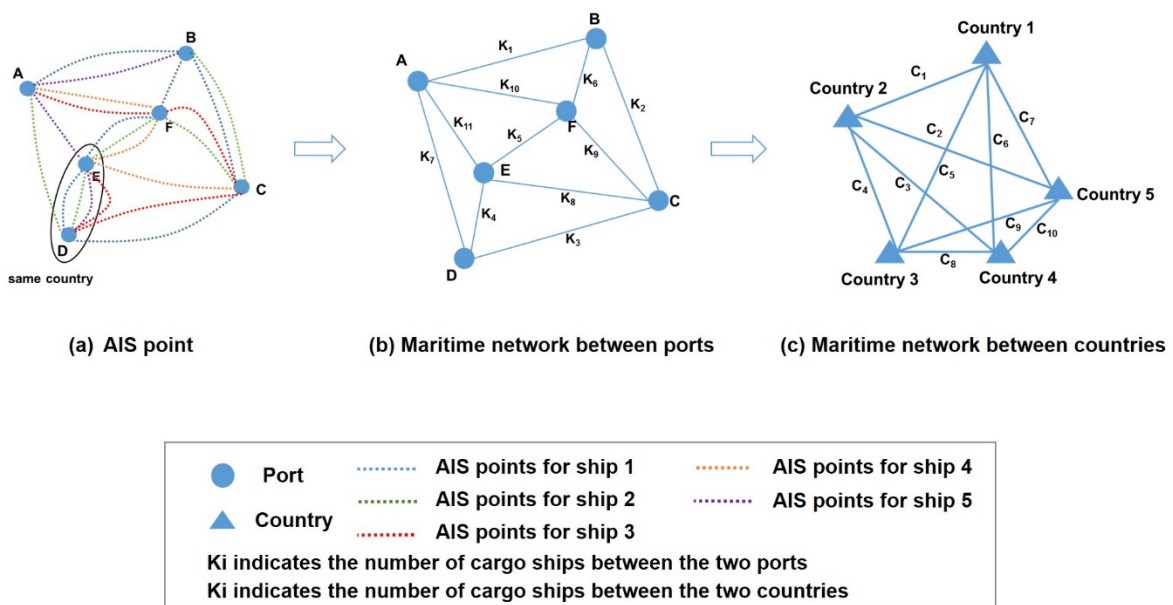

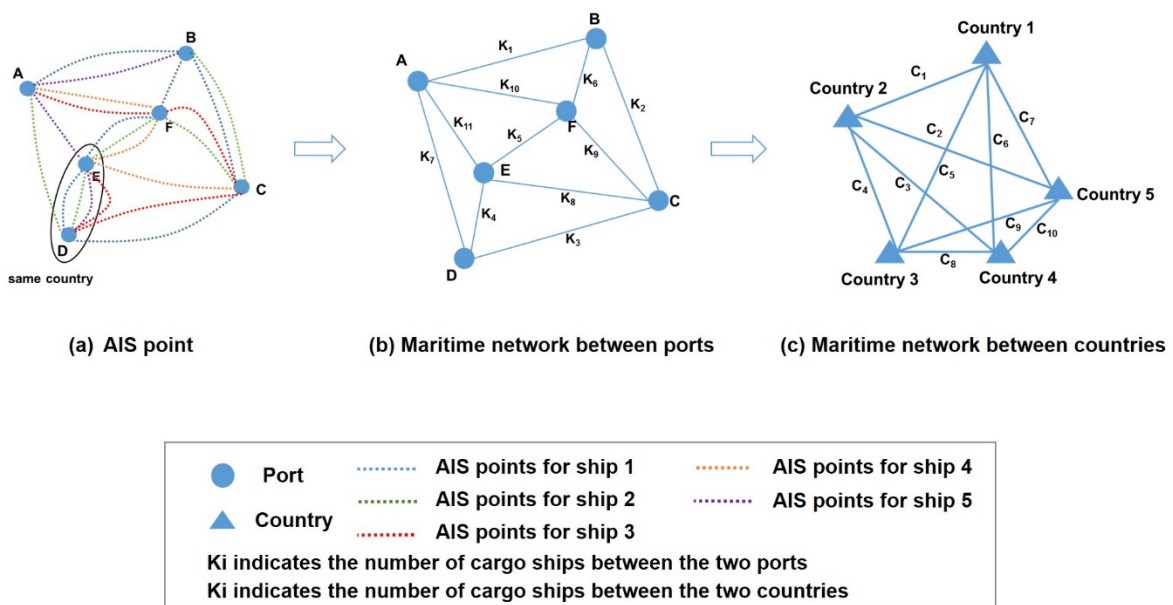

**Figure 2.** Construction of maritime network using AIS data: (**a**) Positioning different ships routes by real-time longitude and latitude information; (**b**) Identifying the ports in routes, and building a maritime network with ports as nodes; (**c**) Combining WPI and national basic information data to build a maritime network with countries as nodes. Source: Adapted from Yu et al. [35].

### 3.3. Research Method

In this study, we explore the Maritime Silk Road traffic inequality on the basis of degree, social network analysis, dominant flow analysis, and traffic sharing ratio. We also examine the dominant regions and study traffic relations among countries in the dominant region using multiple linkage analyses and a competition–synergy model that is based on the multiple linkage theory.

### 3.3.1. Degree

A degree is an important parameter of complex network topology used to describe the direct influence of network nodes. In a directed network, the degree of a node is divided into "in-degree" and "out-degree". In-degree is the sum of the number of times that a node has been the end point of an edge in the graph, reflecting the attraction of the node to other nodes in the network. Out-degree is the sum of the number of times that a node has been the starting point of a graph, which reflects the influence of that node on other nodes in the network. In an undirected network, the degree of a node is the number of adjacent edges directly connected to the node. The formula is as follows:

$$D_i = \sum_{j=1,j\neq i}^{n} L_{ij} \tag{1}$$

where, $D_i$ is the degree of node $i$, $L_{ij}$ is the number of edges between node $i$ and node $j$, and $n$ is the total number of nodes.

### 3.3.2. Social Network Analysis

The relative importance of a region is characterized by its centrality in a network [48], which in turn is reflected by connectivity and mediation. Since we study the centrality of nine regions, we measure connectivity and mediation by using the degree of centrality and flow betweenness centrality, respectively.

(1) Degree centrality. The degree centrality of a region (region A), which can directly reflect the connection capacity of each region in the maritime network, refers to the sum of the edge weights of region A directly connected with other regions.

(2) Flow betweenness centrality. The flow betweenness centrality reflects the degree of control over the traffic flow in each region; this is used to measure the transshipment capacity of the region. The calculation is shown in Equation (2),

$$C_B(i) = \sum_{j<k} \frac{g_{jk}(i)}{g_{jk}} \tag{2}$$

where $C_B(i)$ is the flow betweenness of node $i$, $g_{jk}$ is the number of all possible paths between nodes $j$ and $k$, and $g_{jk}(i)$ is the number of all possible paths between node $j$ and node $k$ that pass through node $i$.

### 3.3.3. Primary Linkage Analysis

Primary linkage analysis (PLA) is a classic network flow analysis method proposed by Nystuen et al. [49] (shown in Figure 3). The core of this method is to determine whether the dominant outflow of any single region points to another region, the dominant flow (the maximum flow) of the dominant region in the network points to a relatively small region, or the main flow of all other subdominant regions points to a relatively large region and retains the main flow of all regions in the network [27]. Thus, this method is used to extract the backbone structure of the network and to simplify complex network relations. The dominant flow analysis is mainly concerned with the dominant flow in the inflow and outflow regions. The role of each region is determined by the total number of inflow dominant flows received. The larger the inflow, the more prominent its role. A relationship between regions can be further defined as dependent or independent. If the maximum outflow from a region flows to a less prominent region, the region is determined to be independent, as shown in Figure 3, where the node f represents an independent region. Conversely, when the maximum outflow of a region is associated with a more important region, the node shows dependence; in the figure we can see that the nodes a, c, and d are dependent on b. The dominant flow analysis method is used to study the dominance and regional relations of the Maritime Silk Road.

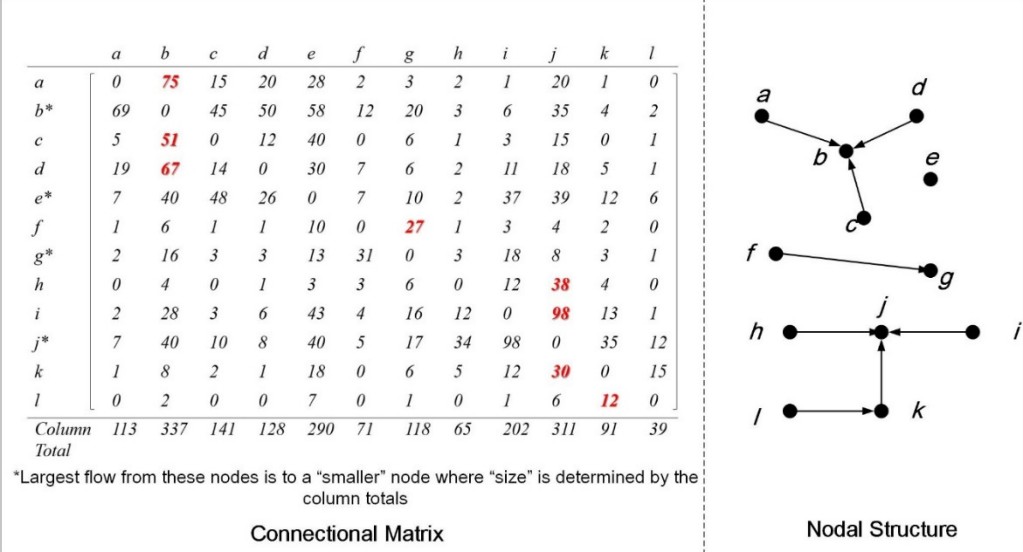

**Figure 3.** Network structure in primary linkage analysis (PLA). The red of the number represents the largest flow of a "smaller" node. Source: Adapted from Nystuen et al. [49].

### 3.3.4. Traffic Sharing Ratio

In order to further measure the dependence of one region on another based on the analysis of regional dominant flows, the traffic sharing ratio, i.e., the ratio of regional dominant maritime traffic to all its external links, is further calculated. The formula is as follows:

$$D_{A \to B} = \frac{\max\limits_{i \in A} W_i}{\sum\limits_{i=1}^{k} W_i} \tag{3}$$

where, $D_{A \to B}$ is the dependence of region $A$ on the maritime traffic flow of region $B$, $\max\limits_{i \in A} W_i$ represents the maximum traffic flow out of region $A$, $k$ is the set of all traffic flows out of region $A$ to other regions, and $D_{A \to B} \in (0, 1]$, when $\max\limits_{i \in A} W_i$ (the maximum traffic flow out of region $A$) accounts for the greater the proportion of $\sum\limits_{i=1}^{k} W_i$ (all traffic flows out of region $A$ to other regions), the closer $D_{A \to B}$ is to 1, which represents the stronger the dependence of region A on the maritime traffic flow of region B.

### 3.3.5. Multiple Linkage Analysis

Multiple linkage analysis, which is an extension of the primary flow analysis method, not only concerns the dominant flow of nodes but also takes into account the flow above a certain threshold; it concerns all the significant connections in the network. In this study, multiple linkage analysis is used to analyze the role of various countries in maritime traffic. The steps are as follows:

First, all traffic flow volumes between countries should be arranged in the order of large ($W_1$) to small ($W_n$), with $W_i$ as the $i$th flow, $i \in n$. $n$ is the set of the total traffic-flow volume between the two countries and $\hat{W}_i$ is defined as the desired flow. The formula is as follows:

Step 1:

$$\hat{W}_1 = \sum_{i=1}^{n} W_i, \hat{W}_2 = \hat{W}_3 = \ldots = \hat{W}_n = 0 \tag{4}$$

Step 2:

$$\hat{W}_1 = \hat{W}_2 = \frac{1}{2}\sum_{i=1}^{n} W_i, \hat{W}_3 = \hat{W}_4 = \ldots = \hat{W}_n = 0 \tag{5}$$

Step $j$: ($j < k$)

$$\hat{W}_1 = \hat{W}_2 = \ldots = \hat{W}_j = \frac{1}{j}\sum_{i=1}^{n} W_i, \hat{W}_j + 1 = \hat{W}_j + 2 = \ldots = \hat{W}_n = 0 \tag{6}$$

Step $k$:

$$\hat{W}_1 = \hat{W}_2 = \ldots = \hat{W}_n = \frac{1}{n}\sum_{i=1}^{n} W_i \tag{7}$$

The set of traffic flow volume expectation $\{\hat{W}_i\}$ represents the spatial structure of the whole flow distribution among countries, and the fitting degree between the discharge expectation value and the real discharge value is measured by calculating the decisive coefficient $r^2$ (Formula (8)) for the result of each step. If the decisive coefficient $r^2$ of step $j$ is the largest, then the maritime traffic flow between the two countries in the top $j$ is the significant flow, i.e., the number of significant flows between countries is $j$.

$$r^2 = 1 - \frac{\sum\limits_{i=1}^{n} \left(W_i - \hat{W}_i\right)^2}{\sum\limits_{i=1}^{n} \left(W_i - \overline{W}\right)^2} \tag{8}$$

(1)    Traffic flow movement and network relations

According to the characteristics of the multiple flow movement, a series of different dispersion modes of the distribution of traffic flow to the country are defined, as shown in Table 1, with the aim of determining the breadth and intensity of the traffic relations between countries in the maritime network.

**Table 1.** Significant flow movement pattern and its influence on network relations.

| Spatial Structure | Description | Node Relations |
|---|---|---|
|  | A significant flow only flows to B. | A is highly dependent on B. |
|  | Significant flows of A and B both flow towards each other. | A and B are complementary and dependent. |
|  | A significant flow flows to more destination nodes. | There is potential competition for A among destination nodes. |
|  | More source nodes significantly flow to B. | B is the hub of the source nodes. |

(2)    Competition–synergy model

The significant flows derived from the multiple linkage analysis method can reflect not only the importance of countries but also the interrelations of the countries [50]. Regardless of the condition of the formation of traffic relations between countries, the distribution of significant traffic flows in the network determines the quantitative judgment on whether there is a relation between competition and synergy to then analyze the current situation of traffic. As shown in Figure 4, in the maritime traffic network, if there are significant flows from more than one country to a target country, the target country plays a more important role in the traffic flow links of these outflow countries, and if multiple target countries receive significant flows from the same country, then these target countries will form competitive relations. For example, if the significant flow of A1 flows to B and C, then B and C will form a competitive relationship. If there is an intersection between the target countries pointed to by a significant flow of two countries, the two countries will form a synergy relationship. For example, when the significant flow of A1 flows to B, C, E, the significant flow of A2 flows to B, C, D, and the target country has an intersection {B, C}, then A1 and A2 have a synergy relationship. Both the "competition" and "synergy" networks between countries are undirected weighted networks. The relationship between competition and synergy has no direction and the two ends are equal, and the degree is divided into strong and weak. The intensity of competition is the number of significant flows received by two target countries from the same country of origin. If two target countries undertake significant flows of the same number of original countries, the competition between the two countries is more intense. The intensity of synergy between the two countries is the intersection of the outflow target countries; the greater the intersection, the greater the synergy intensity. To highlight the main traffic relations between countries, they are classified as follows: (i) a relation with intensity equal to 1 is defined as a "non-significant relation." (In the figure, D and C form a non-significant competitive relation, A2 and A4 have a non-significant synergistic relation); and (ii) a relation with intensity greater than 1 is defined as a "significant relation". In a significant relation, a relation with intensity equal to 2 is defined as a "weak relation", and a relation with intensity greater than 2 is defined as a "strong relation" (In the figure, B and C form a

strong competitive relation; A1 and A3 form a strong synergistic relation; A2 and A3 also form a strong synergistic relation).

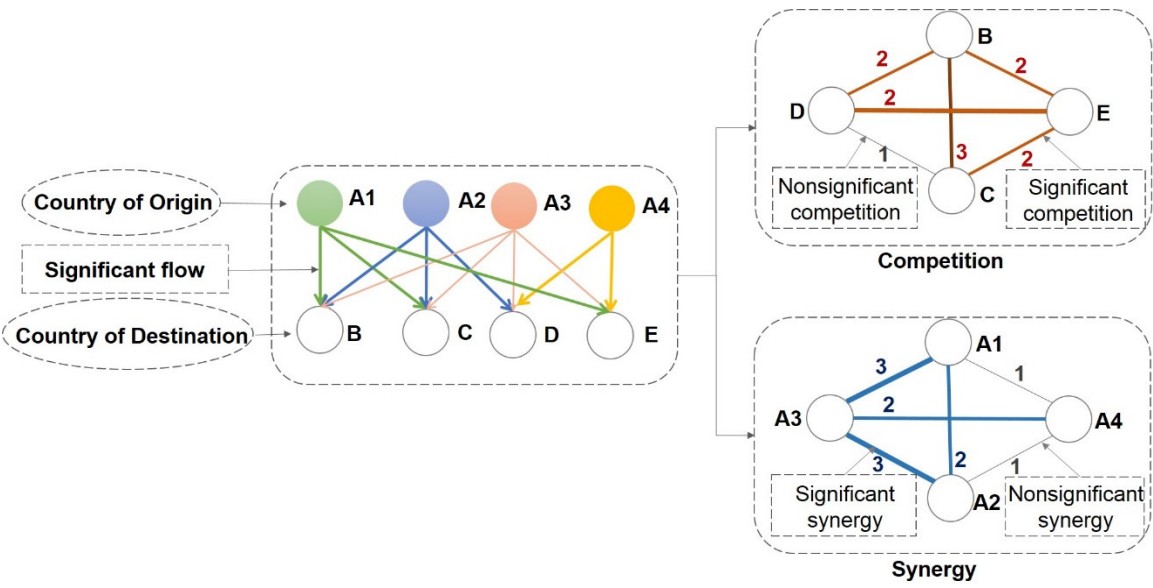

**Figure 4.** Competition–synergy model based on multiple linkage theory. Source: Adapted from Li et al. [50].

## 4. Results

### 4.1. Traffic Inequality Analysis

4.1.1. Spatial Distribution of Traffic Flow

According to the AIS trajectory data, the Maritime Silk Road's shipping topology network is constructed using the traffic volume from the departure country to the target country as the edge weight (shown in Figure 5a). Meanwhile, the traffic volume between different regions is quantified, and the results are shown in Figure 5b. Additionally, Table 2 shows the ratio of the top 20% of the traffic volume to the total traffic volume in the region.

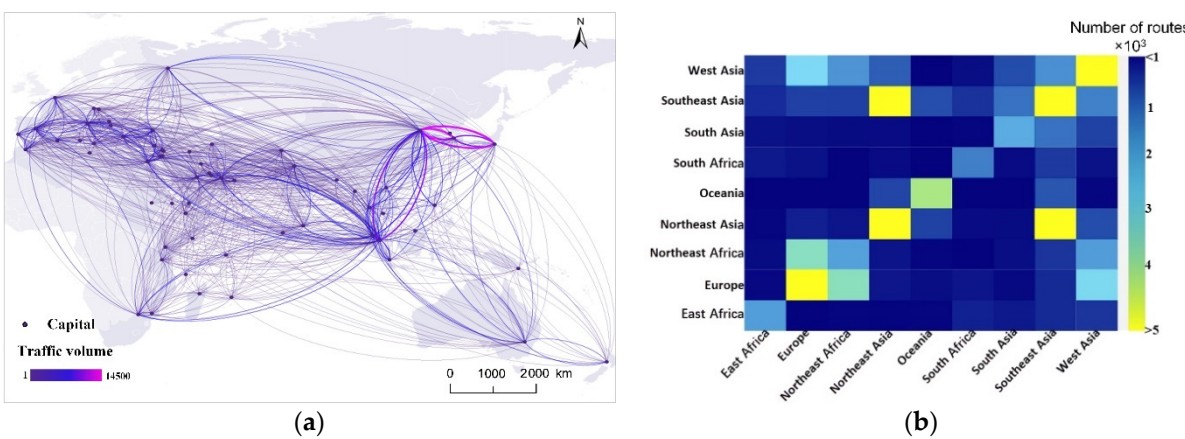

(**a**)           (**b**)

**Figure 5.** Formalization of maritime traffic flow distribution: (**a**) Shipping topological network of maritime traffic flow. The visualization is made by Gephi; (**b**) Thermal map of route distribution. The lighter the color, the greater the maritime traffic volume between the two regions.

**Table 2.** The ratio of top 20% of the traffic volume to total traffic volume in the region.

| Region | Northeast Asia | Southeast Asia | Europe | West Asia | South Asia | Oceania | Northeast Africa | East Africa |
|---|---|---|---|---|---|---|---|---|
| Top 20 (%) | 57.72% | 81.14% | 79.41% | 79.41% | 79.20% | 82.69% | 63.58% | 80.65% |

According to the traffic distribution of the Maritime Silk Road, the maritime traffic shows the characteristics of inequality. From the viewpoint of the overall layout, the distribution of traffic flow in east Asia is the densest, while that in east Africa and Oceania is relatively sparse. From the perspective of the route distribution, intraregional maritime traffic volume is generally greater than interregional traffic volume, which shows that it is more convenient to carry out maritime traffic in the same region and there is a greater probability of forming stable maritime contacts. At the same time, the volume of maritime traffic between west Asia and other regions is relatively high; it is the most active region in the maritime traffic network, followed by southeast Asia. Europe, northeast Africa, northeast Asia, southeast Asia, and Oceania have frequent maritime connections, indicating that geographical proximity plays a key role in their close maritime communication. Finally, from the perspective of the distribution of traffic volume, the top 20% of maritime traffic in each region accounted for more than 57% of the total traffic volume in the region. More than half of the traffic volume was borne by one-fifth of the traffic flow in the region. In Southeast Asia, Oceania, and East Africa, the ratio reached more than 80%.

It can be seen that the distribution of traffic flow on the Maritime Silk Road is extremely unequal, with obvious differences among regions. The denser the distribution of traffic flow, the more active the region's participation in international trade cooperation. We recommend that regions with low participation should actively strengthen maritime contacts with neighboring regions that have high participation to enhance the maritime strength and competitiveness of the region.

### 4.1.2. Regional Dominant Role Analysis

In this analysis, we first measure the maritime connection capacity and the transshipment capacity using the degree centrality and the flow betweenness centrality with UCINET 6, respectively (shown in Figure 6). As seen in Figure 6a, the maritime connection capacity along the Maritime Silk Road varies significantly in space, and the overall pattern is "west weak, east strong". West Asia, northeast Asia, and southeast Asia have the strongest maritime connections with other regions, forming a "triangular core." On the other hand, Oceania, South Asia, East Africa, and South Africa have weaker maritime connections with other regions. Europe, Northeast Africa, and West Asia, which are located near the Mediterranean Sea, perform similarly in terms of the degree of maritime connections, and although they are not as strong as east Asia, they can still have a local advantage.

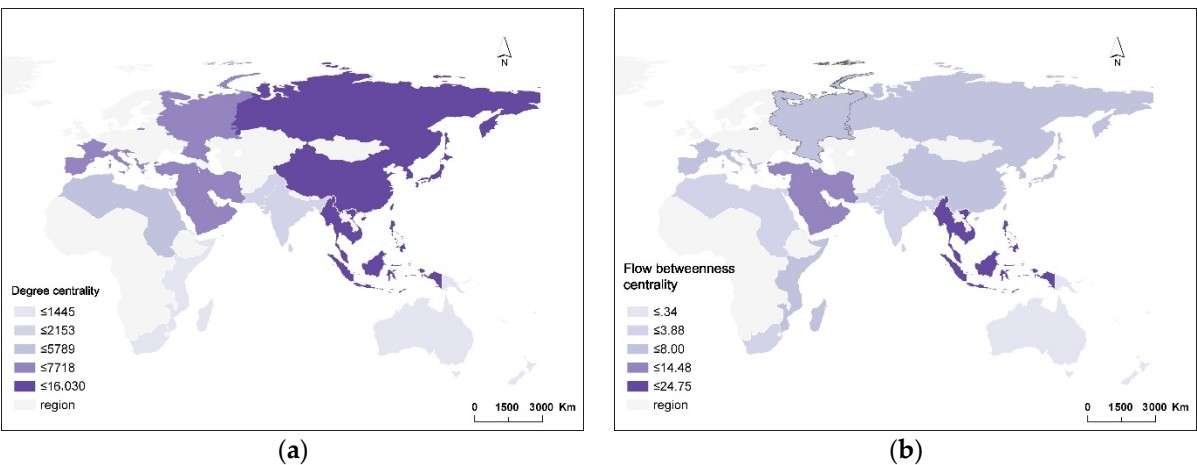

(**a**)      (**b**)

**Figure 6.** Centrality analysis: (**a**) Regional maritime connection capacity; (**b**) Regional maritime transshipment capacity. The value is calculated with UCINET 6.

It can be seen from Figure 6b that the spatial distribution of transshipment capacity along the Maritime Silk Road has a core-edge structure. West Asia and southeast Asia have strong transshipment capabilities; southeast Asia has more obvious advantages. From a geospatial point of view, both are in relatively central positions and have important navigation channels and straits. The transshipment capacities of northeast Asia and Europe are similar, while that of Africa is weaker. Although south Asia is close to the geometric center along the Maritime Silk Road, its transshipment capacity is weak owing to the region's long-term political turmoil and poor economic environment. Oceania has the weakest transshipment capability. Due to its relatively remote and independent geographical location, there are only a few routes available for transshipment through Oceania.

The analysis of regional centrality shows that the west, northeast, and southeast regions of Asia have comparative advantages in maritime connection capacity, while Southeast Asia and West Asia have more prominent transshipment capacities.

On the basis of centrality analysis, and with the help of primary linkage analysis, the status and relations of each region are further studied (shown in Figure 7). It can be seen that the Maritime Silk Road primary traffic flow forms two groups: a western group centered around Europe and an eastern group centered southeast Asia. These groups are relatively independent and show distance proximity characteristics indicating that countries tend to trade closely with their neighboring countries. Among the eastern groups, Southeast Asia is the region that bears the most dominant flow; Northeast Asia, South Asia, South Africa, and Oceania have the largest traffic flow to southeast Asia. The two-way traffic volume between Northeast Asia and Southeast Asia is over 10,000 vessels. At the same time, the regional traffic sharing ratio within this group is relatively high, with 86.4%, 69.4%, 55.4%, 53.7%, and 48.5% in Northeast Asia, Southeast Asia, Oceania, South Asia, and South Africa, respectively. This shows that half of the traffic in these regions often flows to another region. Therefore, Oceania, South Asia, and South Africa are highly dependent on Southeast Asia, while Northeast Asia and Southeast Asia have formed strong marine traffic relations. In the western group, Europe has the most dominant flow; Northeast Africa and West Asia have large dominant flows into Europe. The traffic volume between Northeast Africa and Europe, both of which are located on the coast of the Mediterranean Sea, is approximately 3500 vessels; therefore, the traffic is relatively convenient. The sharing ratio of regional traffic in this group is relatively low, with 34.3%, 35.5%, 49.6%, and 59.6% in East Africa, West Asia, Europe, and Northeast Africa, respectively. It can be seen that the distribution of regional maritime traffic flow in the western group is more balanced than that in the eastern group, and the maritime connection will be more stable. From the above centrality analysis, we can see that West Asia, with its geographical advantages, has an extensive and stable traffic contact and that its transshipment capacity is also prominent. It has a gateway identity, and its position in the western group cannot be ignored.

As shown in the above analysis, we deduce that in the Maritime Silk Road network, Southeast Asia (of the eastern group) occupies the most dominant position, Northeast Asia plays an auxiliary role, and West Asia also plays an important role as a gateway. Europe occupies the dominant position in the western group. Therefore, Southeast Asia, Northeast Asia, Europe, and West Asia are the dominant regions in the Maritime Silk Road.

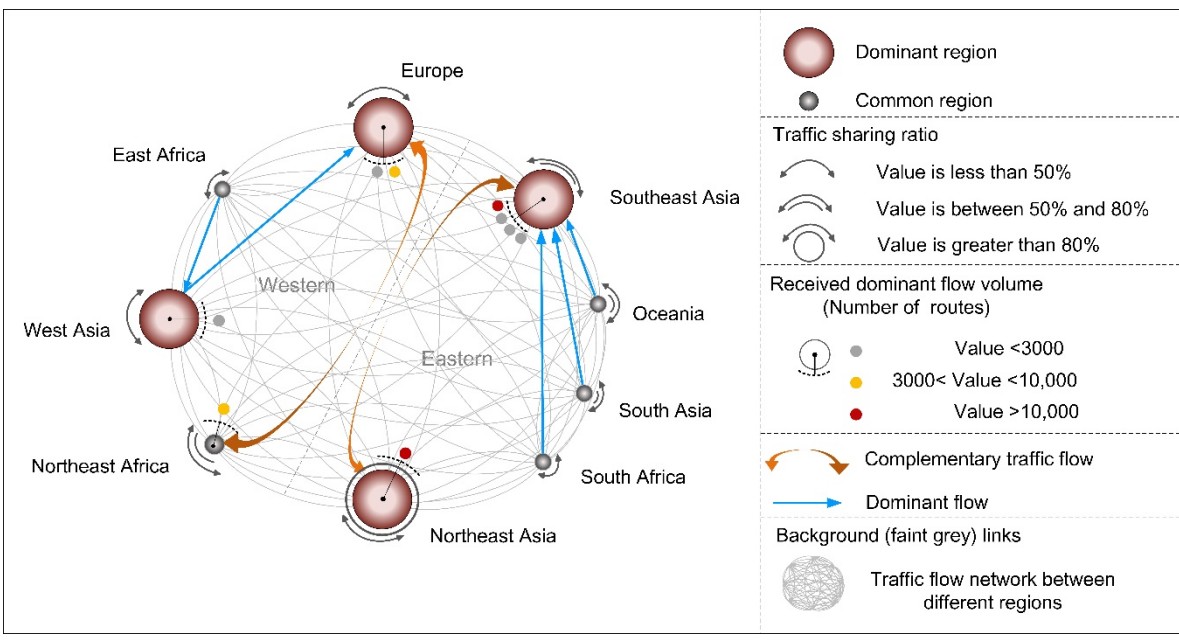

**Figure 7.** Primary linkage analysis. On the basis of traffic flow network between different regions, further analyze the direction and volume of the dominant flows, in order to explore the status and relations of each region.

### 4.2. Traffic Relations Analysis

4.2.1. Role Assessment of National Maritime Traffic

A total of 226,289 OD traffic flows were screened for the four dominant regions (involving 39 countries) as shown in Table 3.

**Table 3.** List of dominant regional countries.

| Southeast Asia | Northeast Asia | West Asia | Europe |
|---|---|---|---|
| Brunei | China | Bahrain | Albania |
| Cambodia | Japan | Cyprus | Croatia |
| Indonesia | Korea | Iran | France |
| Malaysia | South Korea | Iraq | Italy |
| Myanmar | | Israel | Greece |
| Philippines | | Jordan | Malta |
| Singapore | | Kuwait | Montenegro |
| Thailand | | Lebanon | Portugal |
| Vietnam | | Oman | Slovenia |
| | | Qatar | Spain |
| | | Saudi Arabia | Russia |
| | | Syria | |
| | | Turkey | |
| | | United Arab Emirates | |
| | | Yemen | |

Nine countries in southeast Asia, 4 countries in northeast Asia, 15 countries in West Asia, and 11 countries in Europe are considered. The backbone of the Maritime Silk Road network is obtained using the multiple linkage analysis method, and then its role is analyzed along the Maritime Silk Road, as shown in Figure 8. The traffic flow out of West Asia is the most significant, and Europe has received the most significant traffic inflow.

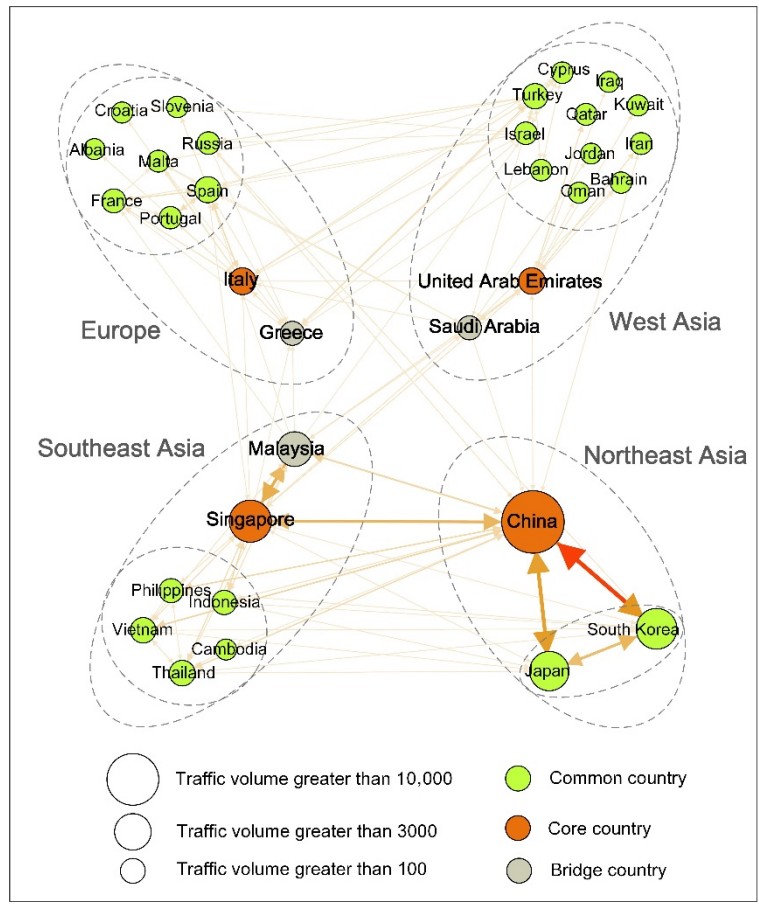

**Figure 8.** Role assessment of countries in the Maritime Silk Road traffic. The visualization is made by Gephi.

From the perspective of significant regional flow, Italy has strong influence over maritime traffic flow in Europe and participates in the significant traffic flows of all countries along the European–Mediterranean region. Eighty percent of Italy's national boundary is a sea boundary, facing the Mediterranean Sea in the east, west, and south. A total of 148 ports are distributed along the nearly 8000 km coastline, of which Genoa is the second largest port in the Mediterranean Sea. With the help of superior maritime geographical conditions and a strong port support, Italy has become a leading "ship distribution center" in the European–Mediterranean region. In West Asia, the UAE has a stronger control over maritime traffic flows in the region than other countries and participates in significant flows in seven countries, including Oman, Bahrain, Qatar, Kuwait, Saudi Arabia, Iraq, and Iran, all located near the Persian Gulf coast. In addition, the UAE has the largest free-trade port in the Middle East: Dubai Port, which is the world's premier transit trade port. In northeast Asia, China undertakes the significant flows of all countries in the region; the number of significant flows undertaken by China is far ahead of the region. As the largest manufacturing country in the world, China has become the distribution center of maritime traffic in not just Northeast Asia but also the rest of the world. Moreover, China's Hong Kong and Shenzhen ports, as well as other large hub ports, are geographically close to the world's largest port—Singapore Port, making it easier to share international maritime traffic resources with it. In southeast Asia, Singapore and Thailand share the same number of significant traffic flows within the region; however, Singapore has a much more significant traffic volume than Thailand. The quantity of significant flow undertaken by a country is correlated to its maritime status to a certain extent. Traffic volume carried by significant flows is also an important factor in describing a country's maritime influence (the greater the traffic volume, the greater the country's maritime influence). It can be

seen that Singapore has a stronger control than Thailand over the maritime traffic flow in Southeast Asia.

From the perspective of interregional significant flows, in the European–Mediterranean region, Greece participates in the largest number of interregional significant flows. Lebanon, Cyprus, and Turkey in West Asia and Singapore and Malaysia in Southeast Asia rely on Greece for traffic flow. By virtue of its proximity to western Asia, Greece has more interregional significant flows than intraregional significant flows; it is an important country for maritime trade in the European–Mediterranean region and West Asia. In West Asia, Turkey bears the most interregional significant flows in the European–Mediterranean region that come from Spain, Malta, Italy, Greece, and Russia. Similarly, it participates in more interregional significant flows than intraregional significant flows, indicating that Turkey has a greater influence on maritime traffic with countries in other regions. In the European–Mediterranean and western Asia region, Greece, Turkey, Cyprus, Lebanon, and Israel form a cross-regional maritime traffic-flow concentration circle, with frequent sea traffic between them. Saudi Arabia has the most balanced distribution of significant flows among the countries that undertake significant interregional flows; all the flows come from countries with high traffic volumes, such as Italy, Spain, and Malta in the European–Mediterranean region and China, Singapore, and Malaysia in East Asia. Therefore, Saudi Arabia plays an important role in maintaining interregional maritime trade. In northeast Asia, China is the country that undertakes the most significant interregional flow and has the widest range. In addition to relying on Southeast Asia, Northeast Asia also has important traffic relations with two maritime powers in west Asia—the UAE, and Saudi Arabia. In southeast Asia, Malaysia has more significant interregional flows, and it is also dependent on many countries with large traffic volumes in the four dominant regions explored in this study: China, South Korea, Japan in Northeast Asia, the UAE and Saudi Arabia in West Asia, and Spain in Europe and the Mediterranean region. It should be pointed out that as a general node country, compared with Core countries which play a core role in the region and Bridge countries which play a role of inter-regional transit hub, Common countries do not show comparative advantages in two aspects, and therefore do not show outstanding support and leading role in the maritime traffic network.

To summarize, Italy, Singapore, and China play a pivotal role in the maritime traffic network of the Maritime Silk Road, not only at the core of the region but also as a transit hub in the entire maritime network. As local hubs, Greece, Turkey, Cyprus, Lebanon, and Israel comprise the main traffic-flow structure in the eastern Mediterranean Sea, which has played a positive role in promoting maritime trade in the Mediterranean region. The United Arab Emirates, as the country with the greatest influence on maritime traffic in west Asia, is more closely connected to east Asia. Saudi Arabia is the country with the most balanced maritime traffic connections among the Maritime Silk Road countries and is the bridge for maritime trade between Europe and Asia. Although the volume of maritime traffic in east Asia is large, it is primarily limited to local traffic circulation. Therefore, east Asia should make full use of its maritime transportation power to enhance the level of transregional maritime traffic.

### 4.2.2. Maritime Traffic Competition-Synergy Analysis

Based on the competition-synergy model, a Maritime Silk Road dominant regional maritime traffic competition and synergy relations network is constructed, and quantitative analysis is carried out based on the number and intensity of the relations, as shown in Figure 9. The distribution of the degree of traffic relations in various countries is shown in Figure 9a,b. The trend lines of the two relations are similar, and the degree value of most countries is above 10. We define countries with degrees between 30 and 35 as high-level countries, between 20 and 30 as middle-level countries, between 10 and 20 as low-level countries. It can be seen that, except for Malaysia, Singapore, and China, the high-level countries are all west Asian or European–Mediterranean countries; the East Asian countries account for half of the middle-level countries; except for Korea, the low-level countries are

West Asian or European–Mediterranean countries. It can be seen that the countries in west Asia and the Mediterranean region are polarized. In contrast, the traffic relations of east Asian countries are more balanced. In addition, Malaysia, Saudi Arabia, Spain, Greece, Italy, and the UAE have the most competitive relations; France, Malaysia, Saudi Arabia, Spain, Singapore, Italy, and the UAE have the most synergistic relations.

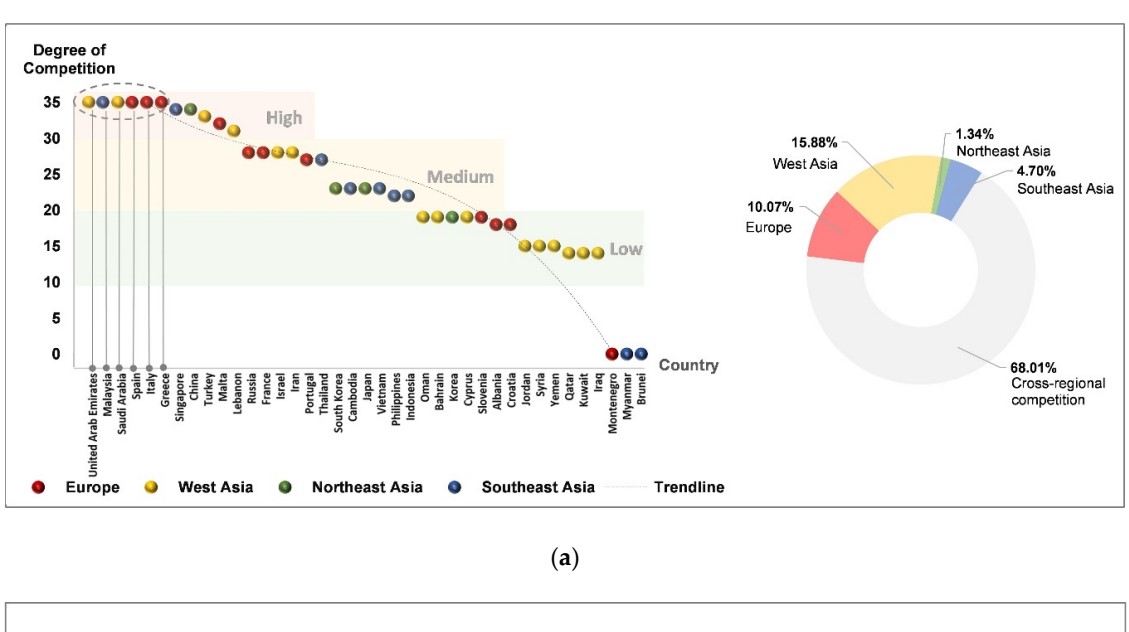

(**a**)

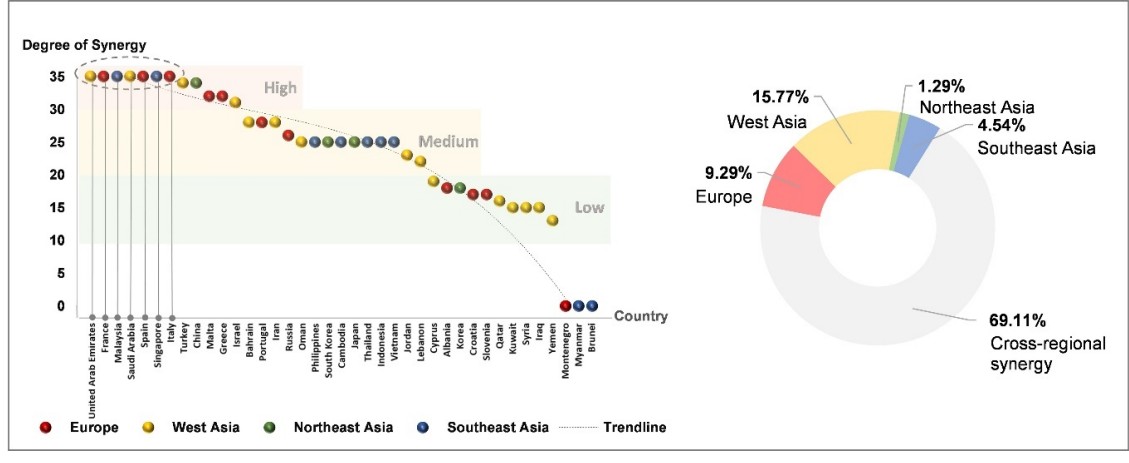

(**b**)

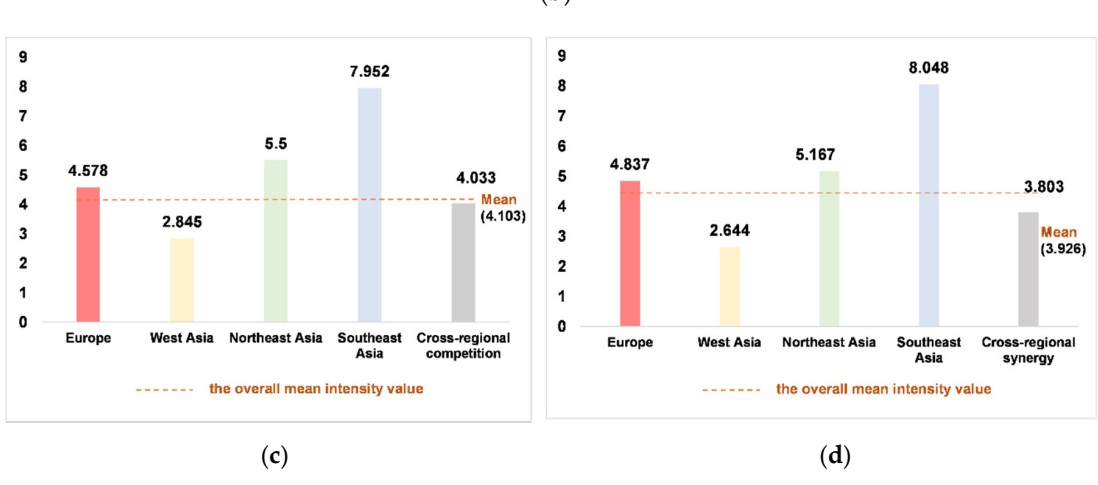

(**c**)                                                       (**d**)

**Figure 9.** Dominant regional traffic relations quantification: (**a**) Degree distribution and percentage of competition; (**b**) Degree distribution and percentage of synergy; (**c**) Competition intensity; (**d**) Synergy intensity.

From the perspective of the spatial distribution and intensity of traffic relations (Figure 9), interregional traffic relations are far more than intraregional traffic relations. Interregional competitive relations account for 68.01%, and interregional synergistic relations account for 69.11%, but the intensity of the relations is similar to the overall mean intensity value, indicating that there are more competition and synergy relations among interregional countries. Within each region, West Asia has the most traffic relations, with competition and synergy relations accounting for 15.88% and 15.77%, respectively, but their intensity is far below the overall mean, indicating that there are more competition and synergy relations among the West Asian countries, but the intensity is very small. The proportions of competition and synergy in Europe are 10.07% and 9.29%, respectively, and their intensity exceeds the overall mean intensity value; the relations between countries are relatively strong. The proportions of competition and synergy among the Southeast Asian countries are 4.70% and 4.54%, respectively; the intensity of their relations far exceeds the overall mean intensity value, and interaction between the countries is extremely strong. The two types of traffic relations in Northeast Asian countries account for less than 2%, but their relations are stronger (second only to Southeast Asia). Therefore, countries in the dominant region have established extensive and robust transregional ocean-going traffic relations, but the intensity of these relations is not very strong. Countries should actively strengthen interregional maritime trade cooperation and build a broader platform for their own development.

This paper explores the characteristics of traffic relations from two perspectives: interregional and regional. Figure 10 depicts the significant traffic relations network between countries across regions. Blue links indicate weak relations and red links indicate strong relations. In the competitive relations network, strong competition accounts for 76.55%, and the synergy relations network accounts for 77.83% of strong synergy, which is significant. As can be seen in Figure 10, countries that undertake more connections in the competitive network tend to have more connections in the synergistic network. The core and edge nodes of the two relational networks are consistent. China, Malaysia, Singapore, and Greece have undertaken the most interregional competition relations, while China, Malaysia, Singapore, and France have undertaken the most interregional synergy relations. The interregional competition and synergy relations of these countries are shown in Figure 11. Further studies will be carried out on these countries. It is evident that China, Malaysia, and Singapore have the strongest competition–synergy relations; Malaysia, Italy, Singapore, the United Arab Emirates, Greece, and Turkey have strong competition intensity, while France, Malaysia, and Singapore have strong synergy. Additionally, it can be seen that there is a high degree of overlap between countries that have a relationship of competition and synergy with core countries. It can be seen that in the interregional maritime network, most countries are both competitive and synergistic, thereby forming competition–synergy traffic relations.

The significant traffic network within each region is shown in Figure 12. It can be seen that the strong relations in Europe are four times as large as the weak relations. Italy, Greece, Spain, France, and Malta have formed the strongest traffic relations network in the European–Mediterranean region. Among them, the competition–synergy relations between Italy and Greece are the strongest, and the competition between Italy and Spain is stronger than others. Slovenia and Albania, which have relatively small traffic volumes, primarily form weak relations with the other eight countries. Although the traffic volume of Malta is also small, it undertakes strong relations, that may be due to the fact that Malta is located at the center of the Mediterranean Sea and adjacent to Italy, a major maritime country. The strong relations in West Asia are twice as large as the weak relations. Six countries, including the UAE, Saudi Arabia, Turkey, Israel, Cyprus, and Lebanon, have formed the strongest traffic network in West Asia. Saudi Arabia and the UAE have the strongest competition–synergy relations, Turkey and Israel have stronger competition with each other, and Turkey and Saudi Arabia have the strongest synergy relations. The significant traffic relations in east Asia are all strong. Northeast Asia, China, South Korea,

and Japan have comparable competition intensity, but the synergy between China and South Korea is more dominant. In southeast Asia, Singapore and Malaysia have formed the strongest competition–synergy relations within the region, Malaysia and Thailand have stronger competition with each other, and Singapore and Thailand have stronger synergies. Overall, the significant traffic network among the countries in dominant regions of the Maritime Silk Road is dominated by strong traffic relations. The geographically adjacent maritime countries are more likely to form the strongest combination of regional competition–synergy.

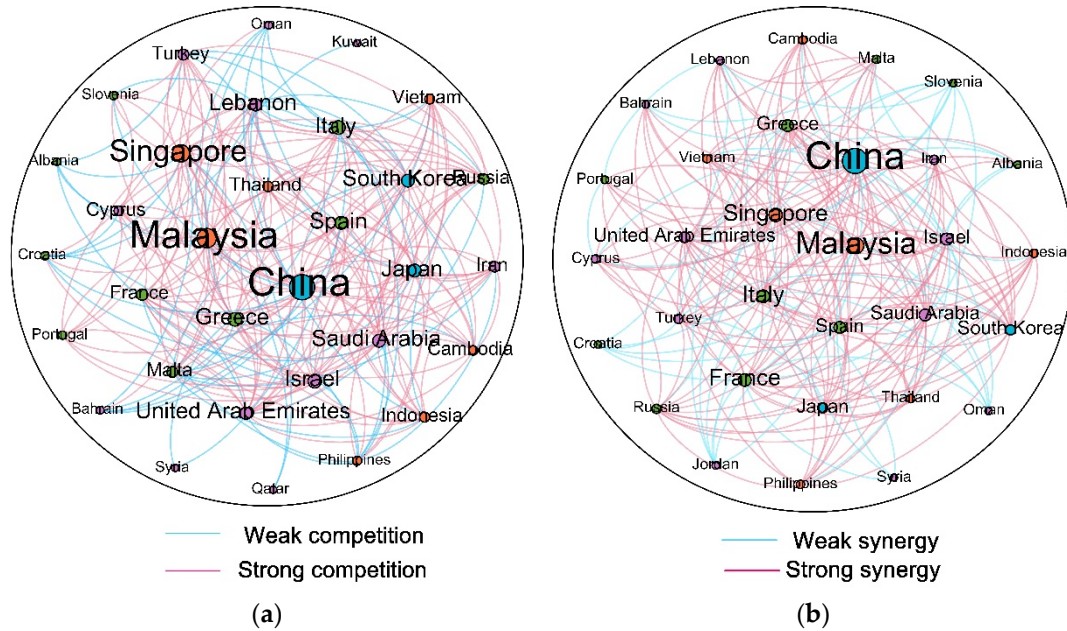

**Figure 10.** Significant traffic network between countries across regions: (**a**) Competition relations between countries across regions; (**b**) Synergy relations between countries across regions. The visualization is made by Gephi.

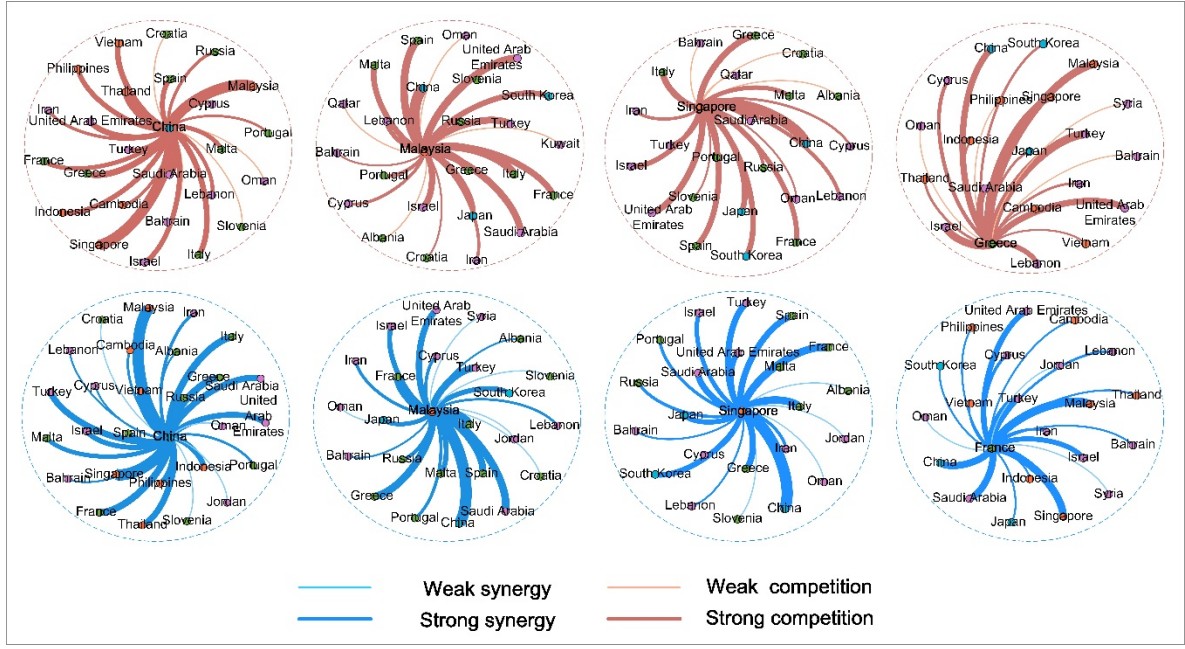

**Figure 11.** Interregional traffic relations in core countries. The visualization is made by Gephi. The thicker the connecting line, the stronger the relation between the two countries.

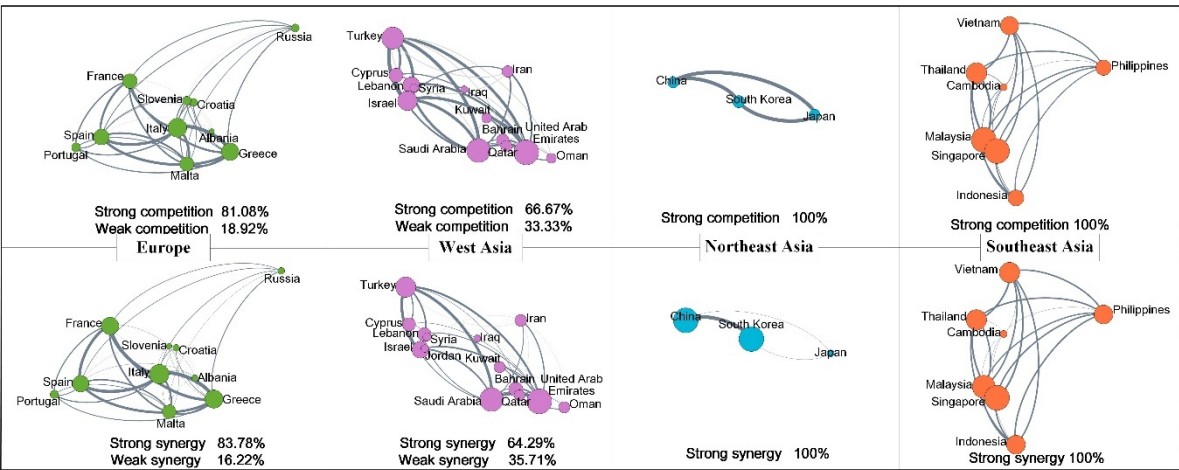

**Figure 12.** Significant traffic network between regional countries. The visualization is made by Gephi. The thicker the connecting line, the stronger the relation between the two countries.

## 5. Discussion

Based on the maritime traffic pattern, this article explores the characteristics of maritime traffic relations in the regions and countries along the Maritime Silk Road so as to provide scientific reference for promoting the positive development of maritime trade and regional economic circulation.

This study found that Europe, West Asia, and East Asia have close maritime traffic exchanges and frequent trade cooperation. They have become the dominant regions of the Maritime Silk Road, participating in 87.8% of the total maritime traffic flow. They have the ability to drive the maritime trade of surrounding areas (such as Africa and South Asia) and the potential to lead the development direction of global maritime trade. As the Belt and Road initiative continues, trade contacts between Europe, West Asia, and East Asia are bound to grow closer. The successful "counter-attack" of the port of Piraeus, which is the second largest port in the Mediterranean, is an example of the Belt and Road initiative promoting common development.

The maritime transshipment capacity of the area along the Maritime Silk Road has a core and peripheral spatial structure, which is highly correlated to the geographical location of the area. Often, the closer the area lies to the center of the geographic space and the closer it is to maritime traffic routes or straits, the more obvious its transshipment advantages. Countries that serve international transshipment can use their powerful and high-quality shipping resources to form a "maritime economic circle" with other powerful countries. With the help of "maritime diplomacy", they can actively integrate themselves into globalization and promote the stable development of the global economy. In addition to geographical factors, political stability, economic prosperity, and maritime safety are important factors in the development of maritime traffic. In recent years, due to the increase in transit fees on the Suez Canal and the threat imposed by the Somali pirates in the canal territory, an increasing number of shipping companies are working to open alternative routes to the Suez Canal (such as the Arctic route and the Cape route) [51]. These actions may also bring opportunities for some south African countries to develop into transshipment centers [28].

Whether or not a country has a strong ability to control maritime traffic flow in a region is often related to the degree of development of its port industry and its geographic location. Studies have proved that in the Mediterranean basin, only 11% of the ports have undertaken traffic volume of more than 10,000 movements/years. The performance of most ports is limited by port capacity. Therefore, it is necessary to reasonably assess globally the performances of a system of ports for potential distribution and re-allocation of traffic, so as to provide a new reference for the development of the port industry [52]. Italy, Singapore, and China have played a pivotal role making the Maritime Silk Road a global maritime

traffic distribution center. Countries with weaker shipping power should actively establish direct or indirect maritime contacts with these major shipping countries, actively integrate into the global maritime trade network, and take advantage of global economic and trade circulation to benefit themselves while also promoting globalization. Greece, Turkey, and Israel play an active role in promoting maritime trade in the Mediterranean region. Taking this as an example, countries should actively build local maritime trade zones, promote economic circulation in large regions with local economic cycles, and further promote global maritime trade. Although the volume of maritime traffic in east Asia is large, it is mainly limited to local maritime trade. East Asia should take advantage of the "Belt and Road" and the "Ice Silk Road" policies, relying on the Arctic waterway to make good use of its strong maritime power and improve the level of ocean trade.

The richer the maritime traffic relations of a country, the more seamlessly the country can integrate into the current maritime trade environment, the greater its international influence, the stronger its ability to participate in regional and even global governance, and the more development opportunities it will obtain. The traffic relations in the areas along the Maritime Silk Road have significant spatial differentiation. The traffic relations in the European–Mediterranean and West Asia regions are polarized, that restricts the overall development of the region. Countries on the periphery of the network, such as Albania and Slovenia, should actively strengthen maritime trade cooperation with their neighboring shipping countries; this is of great significance to the development of the country and the stability of the maritime trade in the region. The traffic relations in east Asia are more balanced, but most countries are in the middle position. To break through the development bottleneck, East Asia should focus on establishing stable maritime contacts with ocean powers, so that an increasing number of countries stand on a higher level of the cooperation platform. We find that most countries have formed a competition–synergy type of maritime traffic relations. From this perspective, the current maritime traffic is already a multidimensional and complex system, and the maritime traffic relations among countries will continue to develop in a multilevel and multidimensional direction. The nations of the world are all part of this vast and complex system, bound together by a common destiny.

## 6. Conclusions

In this study, the maritime traffic network is constructed by using the AIS trajectory data, and the methods of complex network theory, social network analysis, and network flow analysis are used to investigate traffic inequality and relations in the Maritime Silk Road. We explore the dominant regions of the Maritime Silk Road, assess the role of national shipping, provide a more detailed quantitative study of maritime traffic relations, and draw the following conclusions:

(1) The spatial distribution inequality of maritime traffic at the regional and inter-regional levels has caused significant differences in the maritime status and connection characteristics of the regions along the Maritime Silk Road. The maritime connection capacity presents a spatial pattern of "west weak, east strong"; West Asia, Northeast Asia, and Southeast Asia have the strongest maritime connections with other regions, forming a "triangular core" group. The maritime transshipment capacity has a core-periphery structure in space, and the transshipment capacity in West Asia and Southeast Asia is relatively strong. Southeast Asia, Northeast Asia, West Asia, and Europe are the dominant regions of the Maritime Silk Road.

(2) Italy, Singapore, and China have all played extremely important roles in regional and interregional maritime traffic. Greece, Turkey, Cyprus, Lebanon, and Israel form the European Mediterranean–West Asia maritime traffic circle. The UAE has closer maritime contacts with east Asian countries. Saudi Arabia is one of the countries along the Maritime Silk Road that has the most balanced distribution of maritime contacts with other regions and acts like a bridge between Asia and Europe.

(3) From the perspective of the number of traffic relations, the European–Mediterranean and West Asian countries are clearly polarized, while the East Asian countries have a balanced performance. From the perspective of the intensity of traffic relations, the intensity of relations among the southeast Asian countries is the strongest and that of the west Asia countries is the weakest. From the perspective of the type of traffic relations, most countries have formed competition–synergy traffic relations. China, Malaysia, and Singapore have the strongest combination of interregional competition–synergy relations.

We study the maritime traffic status and relations of countries/regions by network flow analysis methods in the "Maritime Silk Road". Our findings can be considered as the first step in the optimization of maritime transportation modes. However, the inaccuracy rate of AIS data is high due to factors such as irregular borders of the port [53]. In addition, AIS data cannot record the ship cargo payload volume. In this sense, AIS data sources have certain limitations. If multi-source big data is introduced for fusion analysis, AIS data can make up for the deficiencies of other data types and give better play to its own advantages.

**Author Contributions:** Conceptualization, Naixia Mou and Haonan Ren; Data curation, Jinhai Chen, Tengfei Yang, and Lingxian Zhang; Formal analysis, Jiqiang Niu and Tengfei Yang; Funding acquisition, Naixia Mou, Jinhai Chen, and Jiqiang Niu; Investigation, Jinhai Chen, Jiqiang Niu, and Tengfei Yang; Methodology, Naixia Mou; Software, Haonan Ren; Validation, Lingxian Zhang and Feng Liu; Visualization, Yunhao Zheng; Writing—original draft, Naixia Mou and Haonan Ren; Writing—review & editing, Yunhao Zheng, Jinhai Chen, and Jiqiang Niu. All authors have read and agreed to the published version of the manuscript.

**Funding:** This research was supported by a grant from Navigation College of Jimei University, National-local Joint Engineering Research Center for Marine Navigation Aids Services, grant number JMCBZD202014, the Major program of National Social Science Fund of China, grant number 20&ZD070, and the Open Fund of Key Laboratory for Synergistic Prevention of Water and Soil Environmental Pollution, grant number KLSPWSEP-A09.

**Institutional Review Board Statement:** Not applicable.

**Informed Consent Statement:** Not applicable.

**Data Availability Statement:** The data presented in this study are available on request from the corresponding author.

**Acknowledgments:** Thanks for three reviewers and all the editors who contributed to their comments and suggestions to improve the quality of the manuscript.

**Conflicts of Interest:** The authors declare that they have no known competing financial interests or personal relationships that could have appeared to influence the work reported in this paper.

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
