# Peer review of "Traffic Inequality and Relations in Maritime Silk Road: A Network Flow Analysis"

_ijgi, doi:10.3390/ijgi10010040_

Round 1

Reviewer 1 Report

If type of vessel (trafic) information was to be included in the analysis, would it bring any addtional value? 

It is unclear how competition is formed in lines 312-314 on page 9. "However, there is competition between different target countries to which the country's significant flow points. For example, if the significant flow of A1 goes to B and C, then B and C form a competitive relationship with A1." It appears that these two sentences are in fact contradictory. The authors should explain this relationship in more detail. This ambiguity is emphasized even more in lines 320 and 321: "The intensity of competition is the number of countries competing for the same number of countries of origin."

Figure 7 is difficult to interpret. Especially the background (faint grey) links that do not carry any information. The authors should try to reformat this figure to be more clear.

The authors should explain in more detail how they measure the "the quality of signifficant flows" in lines 457 and 458: "The quality of the significant flows is also an important factor in describing a country's maritime influence.".

Figure 8 labels some countries as "command", the term that does not appear anywhere in the article text. The meaning of this term should be explained.

Author Response

Dear reviewer,

        Thank you very much for your comments concerning our manuscript entitled “Traffic inequality and relations in Maritime Silk Road:A network flow analysis” (Manuscript ID ijgi-1043244). Those comments are all valuable and very helpful for revising and improving our paper. We have studied your comments carefully and have made corrections that we hope will meet with your approval. The revised portions are marked in Red. The primary corrections in the paper and the response to your comments are in the attached file. Please see the attachment.

With thanks

Best regards

Naixia Mou on behalf of the authors.

Reviewer 2 Report

Into details:

The legend of figure is not corresponding to the various geographic areas identified. You should modify it to create a full correspondence.

In the linkage analysis it should be clarified how the multi-destination erratic trips are considered: the freight flows are in this case overlapped (e.g. A-B + A-C) but the different amounts (to B and C) are not detectable by AIS data. In theory the call in port C could be for loading only (origin) without unloading. Some typical examples are in Mediterranean areas where ship's call from Far East are very often intermediate steps towards Northern Europe Range ports, with negligible unload and majority of load to the next ports. This happens also within North and South Mediterranean countries.

This reality mitigate a lot the leading role of Italy, Greece and Spain discussed in section 4.2.1 because the many calls in their ports are rarely corresponding to unloading of relevant quantity of goods.

In this sense the limit of methodology based on AIS should be recalled.

Moreover, the recommendation at the end of section 4.1.1 is hardly related with the previous results. This sequential connection should be better explained.

The limitation of port capacity, which strongly condition the potential ports' development, particularly but not exclusively in the Mediterranean area, also as a consequence of increasing dimensions of ships, should be considered and discussed (e.g. see Marlow, P.B., Measuring lean ports performance. International Journal of Transport Management, 1, 189–202, 2003.)

Finally, the reference to the pandemic situation should be eliminated or reformulated to specify only if data collection was or not affected by it. 

Author Response

(The authors gave the same response as above.)

Reviewer 3 Report

In this paper, the authors use multiple network-related metrics to study the Maritime Silk Road traffic network. The paper is well written, in my opinion, and the results are valuable from both a scientific and practical perspective.

I only have the following comments, which I consider minor:

  • I guess that in Table 1 (3rd row) it should be “A significant flow flows to more destination modes”.
  • The authors should indicate how the network-related metrics were computed, and if any specific software package was used.
  • Although the graphics and plots provided throughout the paper are great, some of them may be hard to interpret for a reader which is not heavily specialized in network theory. For this reason, I think that the authors should revise the explanations associated with each Figure and possibly expand some of them accordingly. One option could be enriching the descriptions provided in each Figure’s caption.
  • With regard to Figure 9, I do not particularly like the lines plotted in c) and d) (which would be better for a time series plot) and the polynomial fit used in a) and b). Is this polynomial fit meaningful? What does x represent in the polynomial equations displayed?

Author Response

(The authors gave the same response as above.)
